# Effectiveness of Closed System Drug Transfer Devices in Reducing Leakage during Antineoplastic Drugs Compounding

**DOI:** 10.3390/ijerph18157957

**Published:** 2021-07-28

**Authors:** Maria Teresa Piccardo, Alessandra Forlani, Alberto Izzotti

**Affiliations:** 1IRCCS Ospedale Policlinico San Martino, Mutagenesis & Cancer Prevention Unit, 16132 Genova, Italy; alessandra.forlani@hsanmartino.it (A.F.); izzotti@unige.it (A.I.); 2IRCCS Ospedale Policlinico San Martino, Laboratory of Tumor Epigenetics, 16132 Genova, Italy; 3Department of Experimental Medicine (DIMES), University of Genova, 16132 Genova, Italy

**Keywords:** closed system drug transfer device, antineoplastic drugs, drug compounding, gemcitabine, environmental monitoring

## Abstract

This study, conducted in a centralized cytotoxic drug preparation unit, analyzes the effectiveness of two closed system drug transfer devices (CSTDs) in reducing leakage during antineoplastic drug compounding. Wipe/pad samplings inside and outside the preparation area were taken during surveillance programs from 2016 to 2021. All samples were analyzed for gemcitabine (GEM) contamination. In 2016, the presence of GEM in some samples and the contamination of the operators’ gloves in the absence of apparent drug spilling suggested unsealed preparation systems. In subsequent monitoring, GEM was also evaluated in the vial access device and in the access port system to the intravenous therapy bag of Texium^TM^/SmartSite^TM^ and Equashield^®^ II devices after the reconstitution and preparation steps of the drug. The next checks highlighted GEM dispersion after compounding using Texium^TM^/SmartSite^TM^, with positive samples ranging from 9 to 23%. In contrast, gemcitabine was not present at detectable levels in the Equashield^®^ II system in all of the evaluated samples. The Equashield^®^ II closed system seems effectively able to eliminate spills and leakage during gemcitabine compounding. Since drugs with different viscosities can have different effects on CSTDs, Equashield^®^ II needs to be considered with other antineoplastic drugs during a structured surveillance program.

## 1. Introduction

Antineoplastic drugs, also known as cytotoxic or cytostatic drugs, are medications designed to destroy cells that grow rapidly and uncontrollably, preventing them from replicating or growing. Unfortunately, they are non-selective and do not differentiate between malignant and normal cells; it is therefore likely that they can damage healthy tissues, resulting in adverse health effects [1].

Essential for cancer treatment, they also play an important role in hematology. Additionally, they are used to treat rheumatologic diseases, multiple sclerosis, psoriasis, and lupus erythematosus [2]. These drugs are therefore widely used, and the number of preparations and administrations has increased significantly over the years, highlighting the risk associated with occupational exposure [3,4].

The U.S. National Institute for Occupational Safety and Health (NIOSH) has included antineoplastic drugs in their definition of hazardous drugs because they are dangerous chemical agents that are known or suspected to cause adverse effects from exposure in the workplace. It is well known that healthcare workers who are continuously exposed to low doses of antineoplastic drugs may experience acute symptoms such as allergic reactions, headache, nausea, and vomiting or long-term effects including genotoxicity, infertility, and fetal abnormalities [5]. To minimize exposure, the guidelines for the safe handling of antineoplastic drugs and for protecting workers recommend using biological safety cabinets (BSCs) with a laminar vertical airflow hood and external exhaust in preparation areas as well as wearing adequate personal protective equipment (PPE) and undergoing staff education [6]. Wipe sampling for antineoplastic drug surface residue of is considered the method of choice to assess the risk of occupational exposure and to determine the effectiveness of safe handling procedures in healthcare settings [7].

The exposure to antineoplastic drugs can occur via direct and indirect contacts. The main routes of direct exposure are the inhalation of aerosolized drugs, ingestion, and injection through accidental needle sticks. Spills, leaks, and aerosols are often caused by needles or by Luer lock-based needleless connectors. Indirect exposure from dermal absorption is caused by aerosolized antineoplastic drugs that can settle on work surfaces. A possible contamination source is the open barrel of a standard syringe plunger when it comes into contact with the cytotoxic agent during aspiration and remains exposed to the environment once the drug is discharged from the syringe [8].

Many strategies have been deployed to reduce the risk of occupational exposure to dangerous drugs for healthcare professionals, including control devices designed to act as closed systems and preventing exposure through liquid or vapor leakage. These devices, known as closed system drug transfer devices (CSTDs), are defined by NIOSH as transfer devices that mechanically prohibit the escape of hazardous drugs or vapor concentrations from the system and the entry of environmental contaminants into the system. Closed systems, equipped with a mechanism to regulate the differential pressure inside and outside the vial, limit the potential for aerosol generation and, consequently, the exposure of workers.

Since the publication of the NIOSH Alert in 2004 [9], the use of CSTDs for the preparation of hazardous drugs has been encouraged in United States hospitals, and the European Biosafety Network has also began to promote these prevention devices [10]. However, the interest in and the usage of CSTDs significantly increased after the publication of the United States Pharmacopeia (USP) General Chapter (800), “Hazardous Drugs-Handling in Healthcare Settings” [11].

Today, several CSTDs are available on the market. They are designed differently from each other, and they should act to maintain a closed connection between the vial and the syringe or transfer device. There are two primary CSTD device-to-device interface designs that are available today: the needle-free common fluid pathway and the membrane-to-membrane needle pathway [12]. CSTDs with a needle-free common fluid pathway use mating membranes or plastic components that, when they are connected, open a common channel for transferring drugs and vapors, and when they are disconnected, the system is closed and sealed. Membrane-to-membrane needle pathway CSTDs use two adjacent membranes that are engaged by one or more needles for the removal of drugs and vapors and for equalizing pressure. As the system is disengaged, the needles are scrubbed of drug residue by the membranes and is stored securely within the system.

PhaSeal^TM^ from BD Medical (Franklin Lakes, NJ, USA) was the first CSTD approved by the U.S. Food and Drug Administration (FDA) in 1998. Since then, a range of CSTDs have been approved as closed system transfer devices, including ChemoLock^TM^/ChemoClave^TM^ (ICU Medical, San Clemente, CA, USA), Equashield^®^ (Plastmed, Ltd., Tefen, Israel), Equashield^®^ II (Equashield, Port Washington, NY, USA), Texium^TM^ (BD Carefusion, San Diego, CA, USA), OnGuard^®^/Tevadaptor^®^ (B. Braun Medical, Bethlehem, PA, USA), Genie^®^ with Spiros^®^ (ICU Medical, San Clemente, CA, USA), Halo^®^ (Corvida Medical, Eagan, MN, USA), Arisure^®^ (Yukon Medical, Durhan, NC, USA) [13].

Since the introduction of CSTDs in early 2000, numerous studies have demonstrated their effectiveness at decreasing surface contaminations and occupational exposure of healthcare personnel [14,15,16,17,18,19].

The primary purpose of this study was to evaluate the effectiveness of two closed system transfer devices (Texium^TM^/SmartSite^TM^ and Equashield^®^ II) in reducing leakage during antineoplastic drug compounding, which was achieved by surface wipe sampling. The antineoplastic drug gemcitabine (GEM) was measured using surface wipe sampling in the work area, in the vial access device, and in the access port system to an intravenous therapy bag (IV bag) after the reconstitution and drug preparation steps. The performance of different CSTDs was also assessed by comparing the most recent literature data.

## 2. Materials and Methods

### 2.1. Study Design and Sample Collection

This study was conducted in the centralized cytotoxic drug preparation unit of a Genova hospital pharmacy department.

The sterile doses of parental cytotoxic drugs were prepared every day through manual compounding in two class II BSCs with a return air system, located in a negative pressure clean room. The return air was filtered through a high efficiency particulate air (HEPA) filter and a carbon filter. The cytotoxic drugs were distributed to the oncology wards of three hospitals.

Every day four nursing operators prepared the cytotoxic drugs, alternating their work of preparing drugs in the BSC (the first operator) and supporting the work of the preparer (the second operator).

Wipe and pad samples were taken during the surveillance programs from 2016 to 2021. Double monitoring was performed in 2018.

In order to assess the antineoplastic drug exposure assessment of the healthcare workers, 5-fluorouracile, gemcitabine, paclitaxel, and platinum compounds were used as markers.

Beginning in 2017, wipe samplings of the spike adaptor and the access port to the IV bag were performed during gemcitabine preparation. Therefore, the comparison results obtained from gemcitabine monitoring are reported in this study are for the CSTDs only.

Until the end of 2019, the CSTDs used for antineoplastic drug compounding included the system solutions Texium^TM^/SmartSite^TM^ (BD), which were afterwards replaced with the Equashield^®^ II (Equashield).

### 2.2. Standard Practices

According to the national guidelines [20,21], cytotoxic drugs were prepared in a BSC using sterile latex rubber chemoprotective gloves and replacing them every 30 min. According to procedure, disposable gowns, overshoes, and head coverings were required.

Antineoplastic drugs and infusion solution followed this path: from the warehouse, where they were stored, they were transferred to the filter area, and from there they were carried to the clean room through the pass-box. Transport cases were used for all handling.

The BSC work surfaces, side walls, and glass barrier were cleaned with 70% ethanol solution (Farmecol 70, Nuova Farmec) before the workday began. Before starting antineoplastic preparation, absorbent sheets with plastic backing were placed on the shelf of the BSC to contain the dispersion of the drugs in case of accidental spillage. Before dilution, each preparation was wiped at the insert point of the drug with a gauze pad moistened with Farmecol 70.

At the end of the compounding process, each drug was sealed in a plastic bag labeled with the identification of the receiver patient. The plastic bags were placed in a rigid plastic container, and they were transferred out of the clean room through the pass-box. From the antineoplastic drug preparation unit, the drugs were transported directly to the patient-treatment department in a closed bag.

The working surfaces were wiped with Farmecol 70 at the end of the work shift and during the day if necessary. A deep cleaning of the clean room floor and walls was conducted with a cleaner containing chlorex at the end of the workday.

### 2.3. Wipe Sampling and Personal Pad

Wipe sampling allowed the verification of possible drug dispersion on the surfaces while the personal pad enabled assessment of the efficiency of the BSC during working activity.

A predetermined wipe/pad sampling scheme for selected surface areas inside and outside the preparation area was studied and repeated over time. Inside the clean room, sampling locations included work surfaces, airfoils, countertops, and BSC power buttons. Moreover, in the active work area, we also took samples from the worktable, the pen used by the second operator, the floor, the intercom, and various handles. Sampling points outside of the clean room included the worktable, handles, case, the office desk, and the phone. The forearm and chest of the operators were sampled using pads. The gloves were also sampled using wipes.

Wipe samplings were conducted using a paper filter (Whatman ashless, grade 41) wetted with 0.2 mL of Milli-Q deionized water. The sample collection was conducted by wiping in two different directions, from up to down and from left to right [22,23,24].

Similar to the wipe samples, the pads were paper filter (Whatman ashless, grade 41). The nursing staff involved in preparing the drugs wore three pads on the outer surfaces of disposable gowns: on the right and left forearm and on the front of the chest [25].

### 2.4. Sample Extraction

After the wipe and pad samplings, each filter was transferred into a 50 mL polypropylene container to be transported to the laboratory, where it was immediately processed. Each filter was wetted with 4.8 mL of deionized water and extracted by ultrasound for 5 min. The extracted samples were filtered with Millex-GP 0.22 µm (Millipore, Burlington, MA, USA) filters and analyzed using a high-performance liquid chromatography system. All of the operations were performed under a chemical hood.

### 2.5. HPLC Analysis

A total of 100 µl of the sample was injected into the HPLC system 1260 Infinity II (Agilent Technologies, Santa Clara, CA, USA), which was equipped with a variable wavelength UV detector and the software OpenLAB CDS ChemStation (Agilent Technologies, Santa Clara, CA, USA). Separation and quantification of gemcitabine were performed at the wavelength λ: 266 nm using a Raptor FluoroPhenyl column 100 mm × 2.1 mm ID and a particle size of 2.7 µm, equipped with a Raptor FluoroPhenyl EXP guard column cartridge with a 5 mm × 2.1 mm ID and a particle size of 2.7 µm and a mobile phase of methanol/water buffered with 0.02 M ammonium acetate at pH 4.7 (2:98, *v*/*v*) at a flow of 0.5 mL min^−1^. All HPLC-grade solvents were purchased from Merck. Gemcitabine (Accord) 100 mg/mL was used as the calibration standard.

### 2.6. Quality Controls

For each monitoring, blank wipes/pads were extracted and analyzed according to the sample procedure to determine the limit of detection (LOD) and to set the zero concentration for each analytical run. The LOD for GEM, calculated as the average value of the field blanks plus 3 times the standard deviation, was 5 ng/wipe. The limit of quantification (LOQ), defined as 3 × LOD, was 15 ng/wipe. Analyzed blanks were always at background signal levels. The precision level obtained from the triplicate standards of the GEM was 0.6%. Recoveries were performed using 6 wet filters wetted with 10 µL of gemcitabine standard, creating 3 filters at 0.05 µg/wipe and 3 filters at 5 µg/wipe as final concentrations. The recovery filters were extracted and analyzed according to sample procedure, resulted in a level of 98 ± 4%.

### 2.7. Statistical Analysis

The statistical significance of the difference between the data obtained using the TexiumTM/SmartSiteTM in 2016–2018 (*n* = 74) and those obtained using the Equashield® in 2020–2021 (*n* = 38) was tested through a non-parametric Mann–Whitney U test using the software Statview (SAS Institute, Cary, NC, USA).

## 3. Results

Table 1 shows the GEM concentration in wipe/pad samples during the antineoplastic drug monitoring programs from 2016 to 2021.

In 2016, the presence of GEM was found in six of the 35 samples. Contamination was present on the grid and the external border of the BSC with 25 and 22 ng/wipe, respectively, and on the worktable with 43 ng/wipe. High concentrations of GEM (3.8 µg/wipe) were found on the left glove of the first operator in the absence of apparent accidental spillage of drug. The second operator’s forearm and right glove were also slightly contaminated (19 and 15 ng/wipe, respectively). From these results, it was assumed that gemcitabine could derive from unsealed preparation systems.

In successive checks from 2017 to 2021, the spike/vial adaptor access and valve IV bag access port of the closed system devices were monitored during gemcitabine compounding. High levels of GEM were evidenced in wipes of devices in the 2017 and 2018 sampling campaigns, but the drug was below the detection limit (LOD) of 5 ng/wipe in 2020 and 2021 checks.

In 2017, the GEM concentrations were 27.0 and 14.4 µg/wipe in the spike and access port, respectively. The results were also confirmed in two 2018 checks. During the first sampling, 2018(I), GEM concentrations in the spike and the IV bag access port were 206.4 and 3.4 µg/wipe, respectively, while during the second check, 2018(II), GEM concentrations were 431.8 and 17.5 µg/wipe. In 2017, a trace of GEM was found on the right forearm of the first operator (20 ng/wipe). In 2018(I), the right and left gloves of the first operator were strongly GEM contaminated (2.6 and 16.4 µg/wipe, respectively), as was the left glove of the second operator (113 ng/wipe). In the 2018(I) monitoring program, the center and the grid of the cabinet were found to be contaminated by GEM (670 and 184 ng/wipe, respectively) as was the handle of the pass-box (286 ng/wipe), evident signs of a widespread dispersion of the drug. In 2018(II), GEM concentrations were also found in the BSC grid (11.4 µg/wipe) and in its external border (409 ng/wipe). In 2020 and 2021, gemcitabine was not present at detectable levels in any wipe/pad samples. Mann–Whitney U test analysis indicated that the difference between the recorded values for the Texium^TM^/SmartSite^TM^ and Equashield^®^ was significantly different, with a U value of 1,159 and a *p* value = 0.0064.

With these results, the study intends to encourage the use of CSTDs, and if properly designed and used, they offer healthcare professionals advanced protection against potentially hazardous drug exposures.

## 4. Discussion

Environmental monitoring has played an important role in protecting workers from exposure to antineoplastic drugs because it has allowed the identification of the weak points in the working procedures. GEM was detected in all spikes and bag access ports of the closed system solution Texium^TM^/SmartSite^TM^, often producing the drug contamination of the gloves of both preparer and support operator, with consequent dispersion outside the BSC. When using the Texium^TM^/SmartSite^TM^ solution, the percentages of GEM-positive samples ranged from 9 to 23%.

In contrast, GEM was not present at detectable levels in any sample when compounding using the Equashield^®^ II system. As a result, the Equashield^®^ II closed system seemed able to effectively eliminate spills and leakage during antineoplastic drug compounding and, consequently, the surface contaminations in the antineoplastic drug unit.

These results are supported by studies focused on the containment function of CSTDs. Texium^TM^ male Luer and SmartSite^TM^ vented vial access were examined by Jorgenson et al. [26] for their airtightness and leak-proof capacity in both preparation and administration practices. They performed two tests using titanium tetrachloride and fluorescein sodium to simulate the escape of vapor and the contamination of the connections between the vial and the syringe and the between syringe and the access port. The visible presence of titanium smoke in the first test highlighted that the system was not able to prevent vapor escape. In the second test, the presence of fluorescein leaking outside the connections during preparation and administration manipulations demonstrated the potential drug release into the work environment. A successive study, with fluorescein also chosen as the tracer to measure contamination during the preparation of a solution using the Texium^TM^ and SmartSite^TM^ systems, confirmed the same results for the same critical points [27].

In contrast, some studies have shown a percentage decrease of antineoplastic drug detectable levels in surface sampling wipes after the implementation of the Equashield^®^ CSTD. Clark and Sessink [28] demonstrated that when using the Equashield^®^ to prepare and administer chemotherapy drugs, the surface contamination for the evaluated cytotoxic agents, cyclophosphamide and 5-fluorouracile, were eliminated. The Equashield^®^ design with a metal rod as a syringe plunger prevents plunger contamination, as shown by Smith and Szlaczky [29]. The authors evaluated the plungers of BD syringes with the PhaSeal^TM^ CSTD against those of the Equashield^®^ using wipe test sampling after repeated withdrawal and re-injection cycles of cyclophosphamide in order to simulate their repeated use. They found significant cyclophosphamide contamination levels on most PhaSeal^TM^ BD syringes, while the Equashield^®^ syringes remained uncontaminated at undetectable levels. Wilkinson et al. [30] proved that Equashield^®^ was qualified to handle hazardous drugs by using 2-phenoxyethanol as the surrogate for cytotoxic drugs when testing the vapor containment performance of different CSTDs according to the NIOSH protocol [31]. The same authors highlighted that OnGuard^®^/Tevadaptor^®^ and PhaSeal^TM^ also met the acceptance criteria for significantly reducing operator exposure, while ChemoClave^TM^ did not meet these requirements. Forshay et al. [6] evaluated the vapor containment abilities of Equashield^®^ II and five other CSTDs (ChemoClave^TM^, ChemoLock^TM^, OnGuard^®^/Tevadaptor^®^, PhaSeal^TM^, and SmartSite^TM^/VialShield^®^) during the tasks of compounding and administration. The performances were assessed by measuring the vapor release for 70% isopropyl alcohol according to the NIOSH protocol [32]. Among the considered CSTDs, only the Equashield^®^ and PhaSeal^TM^ proved to be adequately close in both tasks. Another recent study compared three different CSTDs (PhaSeal^TM^, ChemoLock^TM^, and Equashield^®^ II) for their adoption into the daily practice of compounding and administration [18]. No statistically significant difference in the compounding efficiency was observed among the three different devices, while in terms of ease of use, PhaSeal^TM^ required more steps than the ChemoLock^TM^ and Equashield^®^ II. In terms of ease of use, it also has been shown in a previous study that the Equashield^®^ system is more readily accepted by operators than the PhaSeal^TM^ [33].

From the abovementioned studies, we can deduce the effectiveness of the Equashield^®^ at ensuring the containment of liquid and/or vapor, but this does not preclude that other CSTDs may be equally effective. The differences among the devices as well as the lack of standard quantitative methods for assessing CSTD performances, as underlined by USP (800), do not facilitate a choice for which the currently available CSTDs would be best suited to the daily practices of hazardous drug compounding and administration. A recent study by Besheer et al. [34] highlighted the need to evaluate the performance aspects of CSTDs to select the best system for their intended use. In this study, four commercially available, but not identified, CSTDs were evaluated for different suppliers in combination with different container-closure systems, different vial sizes and vial types, and different caps. The tests assessed the integrity of the systems by using the helium leak test to measure the force required to assemble the vial adaptor, the presence of particles after pushing the CSTD through the rubber stopper, and the hold-up volume that was not extracted from the vial. The helium container-closure integrity test proved a significant variability among the same CSTDs from a single vendor and among different CSTDs, leading the authors to conclude that CSTDs may not be fully sealed and that there may be leaks.

The other performances evaluated by Besheer et al. [34] could affect drug administration and, even if they do not directly affect the compounding steps covered by our study, they are fundamental for the choice of device. The penetration force seems to depend on the CSTD type, including the rubber stopper puncture force. The presence of significant visible particles contaminating the final product due to stopper coring and shedding depends on the CSTD type that is used as well as the presence of subvisible particles, in particular, silicone oil. The hold-up volume or the volume that cannot be extracted from the vial or that remains in the CSTD components could depend on the vial size, the viscosity of the solution, or the CSTD design—in particular, the spike or needle length and the opening position. The authors concluded by asserting that all of these factors may affect drug administration, causing contamination or leading to a systematic underdosing, therefore affecting the drug efficacy.

In another recent paper, Kulju et al. also examined the hold-up volume, comparing the performances of the PhaSeal^TM^, Texium^TM^/SmartSite^TM^, OnGuard^®^/Tevadaptor^®^, Equashield^®^, ChemoClave^TM^, and ChemoLock^TM^ [35]. The authors established that the different CSTDs contribute to volume loss by using sterile water during simulated processes of drug preparation and subcutaneous administration in different measures. Before testing, the authors assumed that the Luer lock adapter, a component required in all membrane-to-membrane needle pathway CSTDs, could be a potential source of volume loss in 0.5–3.0 mL subcutaneous/intramuscular administrations, due to the presence of a dead space of about 0.1 mL. This hypothesis was not confirmed. In fact, two CSTDs of different design, ChemoClave^TM^, a needle-free closed-fluid pathway, and PhaSeal^TM^, a membrane-to-membrane needle pathway, had the lowest volume losses. All of the other CSTDs had more than twice the mean volume loss of the ChemoClave^TM^ and PhaSeal^TM^.

Solutions with different viscosities might behave differently in a CSTD; therefore, had the authors used hazardous drugs instead of sterile water, the results might have been different. The study also highlighted that the volume loss was independent of the prepared volume. Therefore, volume loss can be significant for administrations below a 3 mL threshold, but it becomes less important as the administration volume increases. During the trials, it was also observed that after the connection between the Texium^TM^ closed male Luer and the needle, multiple drops of fluid escaped from the system and collected inside the needle cap. This confirmed that Texium^TM^ is not suitable for intramuscular and subcutaneous administration, and it is probably for this reason that the operative instructions do not include this use.

Considering the above, we confirmed that the choice of CSTD for hazardous drug compounding and administration is not easy to make. It is possible that different devices must be used depending on the drug type, but these assumptions must be validated.

Limitations of our study include its retrospective nature and the relatively small number of cases.

## 5. Conclusions

CSTDs are important supplemental engineering controls for containing the exposure of healthcare professionals involved in the handling of hazardous drugs.

GEM dispersion was found after compounding with the Texium^TM^/SmartSite^TM^, while the Equashield^®^ appeared to be completely tight and able to eliminate exposure to GEM. However, to understand why drugs with different viscosities may have different effects on the device, it will be important to evaluate the performance of the Equashield^®^ with other antineoplastic drugs during a structured surveillance program.

The high interest in this topic has led to many studies that have mainly focused on the containment features of CSTDs; however, it will be important to also verify the functionality attributes of CSTDs as well as their impact on final product quality. It is commonly acknowledged that an important goal is to harmonize testing procedures to undertake real comparisons among studies.

## Figures and Tables

**Table 1 ijerph-18-07957-t001:** Results of GEM concentrations (ng/wipe) in wipe/pad samples during the monitoring programs from 2016 to 2021.

		2016	2017	2018(I)	2018(II)	2020	2021
CSTD Used		Texium^TM^/SmartSite^TM^				Equashield^®^	
INSIDE CLEAN ROOM							
BSC	1 left side	<5	<5	<5	<5	<5	<5
	2 right side	<5	<5	<5	<5	<5	<5
	3 center (cloth)	<5	<5	670	<5	<5	<5
	4 grid	25	<5	184	11,430	<5	<5
	5 external border	22	<5	<5	409	<5	<5
	6 bottom	<5	<5	<5	<5	<5	<5
	7 countertop	<5	<5	<5	<5	<5	<5
	8 protective glass	<5	<5	<5	<5	<5	<5
	9 power buttons	<5	<5	<5	<5	<5	<5
	10 spike	-	27,000	206,427	431,792	<5	<5
	11 fitting Luer lock	-	14,400	3389	17,504	<5	<5
	12 worktable	<5	<5	<5	<5	<5	<5
	13 floor	<5	<5	<5	<5	<5	<5
	14 door handle “inside”	<5	<5	<5	<5	<5	<5
	15 refrigerator handle	<5	<5	<5	<5	<5	<5
	16 drugs cabinet	<5	<5	<5	<5	<5	<5
	17 pass-box handle	<5	<5	286	<5	<5	<5
	18 intercom	<5	<5	<5	<5	<5	<5
	19 pen	<5	<5	<5	<5	<5	<5
NURSING STAFF	20 right forearm operator 1	<5	20	<5	<5	<5	<5
	21 left forearm operator 1	<5	<5	<5	<5	<5	<5
	22 chest operator 1	<5	<5	<5	<5	<5	<5
	23 glove fingers right operator 1	<5	<5	2592	<5	<5	<5
	24 glove fingers left operator 1	3798	<5	16,383	<5	<5	<5
	25 right forearm operator 2	19	<5	<5	<5	<5	<5
	26 left forearm operator 2	<5	<5	<5	<5	<5	<5
	27 glove fingers right operator 2	15	<5	<5	<5	<5	<5
	28 glove fingers left operator 2	<5	<5	113	<5	<5	<5
OUTSIDE CLEAN ROOM							
	29 door handle “outside”	<5	<5	<5	<5	<5	<5
	30 worktable top	43	<5	<5	<5	<5	<5
	31 worktable side	<5	<5	<5	<5	<5	<5
	32 transport case	<5	<5	<5	<5	<5	<5
	33 drug warehouse	<5	<5	<5	<5	<5	<5
	34 phone executive office	<5	<5	<5	<5	<5	<5
	35 desk of nurse coordinator	<5	<5	<5	<5	<5	<5
Positive samples Percentage positive samples		6 17%	3 9%	8 23%	4 11%	0 -	0 -

Note: LoD = 5 ng/wipe; Recovery = 98 ± 4%.

## Data Availability

All data are contained within this manuscript.

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
