# Peer review of "Effectiveness of Closed System Drug Transfer Devices in Reducing Leakage during Antineoplastic Drugs Compounding"

_ijerph, 2021, doi:10.3390/ijerph18157957_

Round 1

Reviewer 1 Report

In this manuscript entitled "Effectiveness of closed system-drug transfer device in reducing 2 leakage during antineoplastic drugs compounding" the authors proposed the evaluation of the effectiveness of two closed system-drug transfer devices (TexiumTM/SmartSiteTM and 96 Equashield® II) in the reduction of leakage during antineoplastic drug compounding. The evaluation was centered on gemcitabine contamination.  The importance of this manuscript is related to the fact that antineoplastic drugs are no selective between normal and cancerous cells. Including, It is known that they can cause a lot of adverses effects when exposed in the workplace.  In this way, a lot of strategy has been performed in order to reduce or eradicate this problem.
In the statistical evaluation, authors revealed that  Equashield® II system was effective in the reduction of the leakage. On the other hand,  TexiumTM/SmartSiteTM did not exhibit effectivity. 
I considered that this manuscript presents good results that support its acceptance in the IJERPH. Additionally, the results are clear and supports all discussion performed. 

Author Response

We thank the Reviewer for comments and constructive criticism.

Reviewer 2 Report

The manuscript presents a quite interesting topic and deserves publication, however, I recommend careful language editing. Additionally, the analytical method should be described in more details (precision, accuracy, LOD, LOQ, quality control, quality assurance should be described).

Author Response

We thank the Reviewer for comments and constructive criticism.

In the revised version, the following changes have been performed according the Reviewer suggestions:

  • The manuscript has been edited for English (The English editing certificate is enclosed).

  • Quality controls of analytical method have been detailed, (paragraph 2.6, Lines 182-191):

2.6. Quality Controls

For each monitoring, blank wipes/pads were extracted and analyzed according to the sample procedure to determine the limit of detection (LOD) and to set the zero concentration for each analytical run. The LOD for GEM, calculated as the average value of the field blanks plus 3 times the standard deviation, was 5 ng/wipe. The limit of quantification (LOQ), defined as 3 × LOD, was 15 ng/wipe. Analyzed blanks were always at background signal levels. The precision level obtained from the triplicate standards of GEM was 0.6%. Recoveries were performed by 6 wet filters wetted with 10 µl of gemcitabine standard for obtaining 3 filters at 0.05 µg/wipe and 3 filters at 5 µg/wipe, as final concentrations. The recovery filters, extracted and analyzed according to sample procedure, resulted in a 98 ± 4% level.

Reviewer 3 Report

The title cover the aims of the study. It is better if the author explain the novelty of the study.

The result section: The authors should ensure that they write a paragraph in order to explain how these results are novel or important to the
field at large. In table 1 should be above the years- 2016-2018 the name of TexiumTM/SmartSiteTM and above years- 2019-2020 the name of Equashield® II).  In addition, in my opinion, there is no statistical analysis of the obtained results on the basis of which it could be concluded whether the obtained data are statistically significant 

The discussion section correlates with the results. Additionally, expanding on the limitations would be a welcome change.

The conclusion section veers away from the main objective, please revise the conclusion so that it reflects the main objective.

In addition, in my opinion, as an expert in health technology assessment, it would be worth conducting an analysis of the usefulness of a given technology to increase the value and innovation of work. Maybe it is worth comparing the purchase costs of the compared closed system transfer devices (TexiumTM / SmartSiteTM and 96 Equashield® II). 

Author Response

We thank the Reviewer for your comments and constructive criticism.

In the revised version, the following changes have been performed according the Reviewer suggestions:

  • The importance of the study has been explained in result section with the following sentence (Paragraph 3, Lines 229-231):

“With these results, the study intends to encourage the use of CSTDs, because properly designed and used, they offer to the healthcare professional an advanced protection against potentially hazardous drugs exposures.”

  • The table has been corrected, adding a row with “CSTD used” in the years 2016-2018 and 2020-2021.

  • The statistical analysis has been added in Methods (Paragraph 2.7, Lines 194-197):

“2.7 Statistical Analysis

The statistical significance of the difference between data obtianed using TexiumTM/SmartSiteTM in 2016-2018 (n=74) and those obtained using Equashield® in 2020-21 (n=38) was tested by non-parametric Mann-Whitney U test using the softwaer Statview (SAS Institute, Cary, NC, USA).”

  • Results of statistical analysis are reported in the Results (Paragraph 3, Lines 226-228):

“Mann-Whitney U test analysis indicated that the difference of the recorded values between TexiumTM/SmartSiteTM and Equashield® was significantly different with a U value of 1,159 and a P value = 0.0064.”

  • Limitations have been added in Discussions (Paragraph 4, Lines 342-343):

“Limitations of our study include its retrospective nature and the relatively small number of cases.”

  • Conclusion has been corrected to reflect the effectiveness of the used CSTDs in reducing leakage of antineoplastic drugs during compounding, with the following sentence (Lines: 348-349):

“GEM dispersion was found after compounding with TexiumTM/SmartSiteTM, while Equashield® appeared to be completely tight and able to eliminate exposure to GEM.”

  • The purchase costs of CSTDs are not analysed in this study as suggested by a reviewer; it is our opinion that the comparison of the devices purchase costs it is not enough for appreciable economic results, rather the entire cost of the processes of preparation and administration should be evaluated. However, the reviewer's valuable advice may be further investigated in a next publication.